# Influence of Micro-Furrow Depth and Bottom Width on Surface Water Flow and Irrigation Performance in the North China Plain

Songmei Zai [1,2], Xuefang Feng [1,2], Donglin Wang [1], Yan Zhang [1] and Feng Wu [2,3,*]

1 College of Water Conservancy, North China University of Water Resources and Electric Power, Zhengzhou 450046, China
2 Henan Key Laboratory of Water-saving Agriculture, Zhengzhou 450046, China
3 College of Water Resources, North China University of Water Resources and Electric Power, Zhengzhou 450046, China
* Correspondence: wufeng@ncwu.edu.cn

**Abstract:** Improving traditional surface irrigation technology and vigorously promoting water-saving surface irrigation are important ways to improve the efficiency of water resource utilization. In our study, we propose a new technology of surface irrigation, micro-furrow irrigation, which combines the advantages of furrow irrigation and border irrigation. The objective of this experiment was to evaluate the effects of micro-furrow depth and bottom width on surface water flow and irrigation performance. Field experiments were conducted from 2019 to 2020 in Zhengzhou City, northern China. This work designed three bottom widths, BW1 (18 cm), BW2 (12 cm), and BW3 (6 cm), respectively, and three depths, D1 (15 cm), D2 (10 cm), and D3 (5 cm), respectively. Moreover, border irrigation was set as control treatment (CK). Additionally, field experiments were validated and simulated using the WinSRFR 5.1 model (Arid-Land Agricultural Research Center, USA). The results showed a significant negative correlation between depth and advance time and between depth and recession time. However, no significant correlation was found between bottom width and advance time, nor between bottom width and recession time. The advance times of micro-furrow irrigation were 1.23–4.77 min less than those of border irrigation. Concerning irrigation performance, compared to that of border irrigation, the application efficiency and distribution uniformity increased by 8–30% and −5–18%, respectively. However, the requirement efficiency decreased by 0–40%. Compared to that of border irrigation, the irrigation quota increased 21.61% under BW3D3 but decreased by 10.46–57.94% under other treatments. Therefore, micro-furrow irrigation can meet irrigation requirements despite low irrigation quota. Comprehensively considering the advance time, recession time, irrigation performance, and irrigation quota, we recommend a micro-furrow shape with a depth of 10 cm or 15 cm and bottom width of 6 cm.

**Keywords:** micro-furrow irrigation; surface water flow; advance time; recession time; irrigation performance

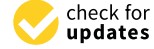



## 1. Introduction

Owing to the increasing population, ensuring food security has become a primary issue for global sustainable development [1]. Recently, global food security has further deteriorated due to the impacts of the COVID-19 pandemic [2]. According to The State of Food Security and Nutrition in the World 2020 by the Food and Agriculture Organization of the United Nations (FAO), approximately 720–811 million people in the world were suffering from hunger in 2020, and nearly one third of the global population did not have access to sufficient food [3]. Moreover, according to The State of Food and Agriculture 2020, the global population has increased rapidly over the past 20 years, whereas shortages of water resources have increased by more than 20%. More than 1.2 billion people in agricultural areas worldwide face serious water resource pressure and drought [4]. Additionally, 11% of farmland suffers from repeated droughts, and more than 60% of irrigated farmland is under

huge water resource pressure [4]. This disparity between food demand and agricultural water supply is a worldwide problem. Thus, the shortage of water resources has become the main restrictive factor for achieving global food security [5,6].

China is the most populous country, largest grain producer, and third largest grain exporter worldwide. It houses approximately 10% of the global farmland to produce nearly 1/4 of the global food supply, which feeds approximately 20% of the global population. Therefore, ensuring food security in China is vital for global food security. Although China is rich in overall water resources, its per capita share is low, approximately one quarter of the world average [7]. Owing to rapid population growth and climate change, the imbalance between supply and demand for water resources in China is becoming increasingly serious. To ensure national water security, China proposed setting up a water conservancy, which prioritizes of water conservation. Agriculture is the most important sector for water consumption. In 2020, agricultural water consumption was 361.24 billion m$^3$, which was 62.1% of the total annual water consumption [8]. However, the present efficiency of water resource utilization in agriculture remains low, at 0.56, far from that of other developed countries. Therefore, agriculture has a great potential for water conservation and requires improvements in the efficiency of water resource utilization.

Improving water-saving irrigation technologies is an important method for increasing the efficiency of water resource utilization [9]. Examples of these include use of ultrasonic sensors [10–12], communication networks and remotely sensed data [13]. By improving the level of field management, the dual purposes of increasing grain yield and reducing water consumption by irrigation can be realized. There are three types of irrigation systems: surface, subsurface, and pressurized irrigation systems. Compared with other irrigation methods, surface irrigation remains widely used worldwide because of its simple field engineering facilities, easy implementation, low energy consumption, and low cost. It services approximately 75% of the total irrigated lands worldwide [14,15] and approximately 86.4% of those in China [16]. In the arid and semi-arid areas of northern China, border irrigation and furrow irrigation are the main irrigation methods [17]. However, they have problems, such as low water application efficiency, low uniformity, and high deep percolation [18–21]. Unlike sprinkler irrigation and micro irrigation, each point along the longitudinal direction in the irrigation unit does not begin and stop receiving water at the same time, resulting in different opportunity times. This manifests as higher infiltration in the front and lower infiltration in the tail and is mainly related to the advance and recession times of the water flow.

For border irrigation systems, the advance and recession times are determined by the factors, such as border dimensions, border slope, inflow rate, cut-off time, microtopography, Manning's roughness coefficient, and soil infiltration properties [22,23]. Thus, researchers have improved border irrigation performance by optimizing the border dimensions [24–27], inflow rate [28,29], cut-off time [30–32], and microtopography [33–36]. Moreover, researchers have improved border irrigation performance by considering temporal and spatial variation in infiltration and roughness [37–42]. Nevertheless, although the performance of border irrigation has been improved, the problem of soil hardening after irrigation remains, which hinders nutrient transportation and crop root respiration.

Unlike border irrigation systems, furrow irrigation systems utilize two-dimensional infiltration. Owing to a concentrated water flow and small wet perimeter, the water flow advances rapidly. This irrigation method causes minimal damage to the soil structure and can keep the surface loose. Its performance depends on the furrow length, furrow shape (i.e., depth, bottom width, and top width), furrow slope, inflow rate, cut-off time, and soil infiltration parameters [43–46]. Field [20,47–49] and simulated experiments (i.e., indoor simulated irrigation, the WinSRFR model, and the Hydrus model) [41,48,50] have been performed to analyze the influence of different field factors on furrow irrigation and thereby determine the best combination of furrow irrigation factors [51,52]. At present, research on furrow irrigation systems has been matured. However, because the width

and depth of the furrows are more than 20 cm, furrow irrigation is prone to seepage and percolation losses, and cannot be used with dense wheat crops.

To remedy these problems, we propose a new technology of surface irrigation, that is, micro-furrow irrigation, which combines the advantages of furrow irrigation and border irrigation. At present, the effects of micro-furrow shapes on water flow characteristics and irrigation performance are unclear. Therefore, the main objectives of this study were:

(1)   to analyze the effects of bottom width and depth of micro-furrows on advance and recession times;
(2)   to evaluate the spatial and profile distribution of soil water and irrigation performance; and
(3)   to determine the optimal combination of bottom width and depth of micro-furrow to maximize irrigation performance.

## 2. Materials and Methods

### 2.1. Field Experiment

Field experiments were conducted from 2019 to November 2020 in the test field at the Henan Key Laboratory of Water Saving Agriculture, North China University of Water Resources and Electric Power, Zhengzhou City, Henan Province, North China Plain (34°72′ N, 113°65′ E, 110.4 m above sea level). The length and width of the experimental field were 48 m and 40 m, respectively. At a depth of 0–60 cm, the soil was a sandy loam and had a bulk density of 1.35 g cm$^{-3}$. The water source was shallow groundwater, which was transported to the field through pipelines.

The field experiments of micro-furrow irrigation were designed with bottom widths of 18 cm (BW1), 12 cm (BW2), and 6 cm (BW3) and depths of 15 cm (D1), 10 cm (D2), and 5 cm (D3) (Table 1 and Figure 1). All experiments were performed using micro-furrows of 30-m length spaced 0.60 m apart with closed boundary ends. The field experiment of border irrigation was control treatment (CK). There were ten treatments and each treatment had three replicates. Ten plots were prepared for the treatments. To reduce the impacts of different plots, each plot was separated by an 80 cm-wide border. The micro-furrows ware formed using mechanized trenching and pressing, and the internal friction angle of their slope was 20°. No crops were planted in the experimental field during the experiments.

**Table 1.** Details of irrigation treatments.

| Treatments | Bottom Width of Micro-Furrow (cm) | Depth of Micro-Furrow (cm) | Field Width (m) | Field Length (m) | Field Slope (m m$^{-1}$) | Inflow Rate of Unit Width (L s$^{-1}$ m$^{-1}$) |
|---|---|---|---|---|---|---|
| BW1D1 | 18 | 15 | 1.8 | 30 | 0.0018 | 2.5 |
| BW1D2 | 18 | 10 | 1.8 | 30 | 0.0018 | 2.5 |
| BW1D3 | 18 | 5 | 1.8 | 30 | 0.0018 | 2.5 |
| BW2D1 | 12 | 15 | 1.8 | 30 | 0.0018 | 2.5 |
| BW2D2 | 12 | 10 | 1.8 | 30 | 0.0018 | 2.5 |
| BW2D3 | 12 | 5 | 1.8 | 30 | 0.0018 | 2.5 |
| BW3D1 | 6 | 15 | 1.8 | 30 | 0.0018 | 2.5 |
| BW3D2 | 6 | 10 | 1.8 | 30 | 0.0018 | 2.5 |
| BW3D3 | 6 | 5 | 1.8 | 30 | 0.0018 | 2.5 |
| CK | - | - | 2.0 | 30 | 0.0018 | 2.5 |

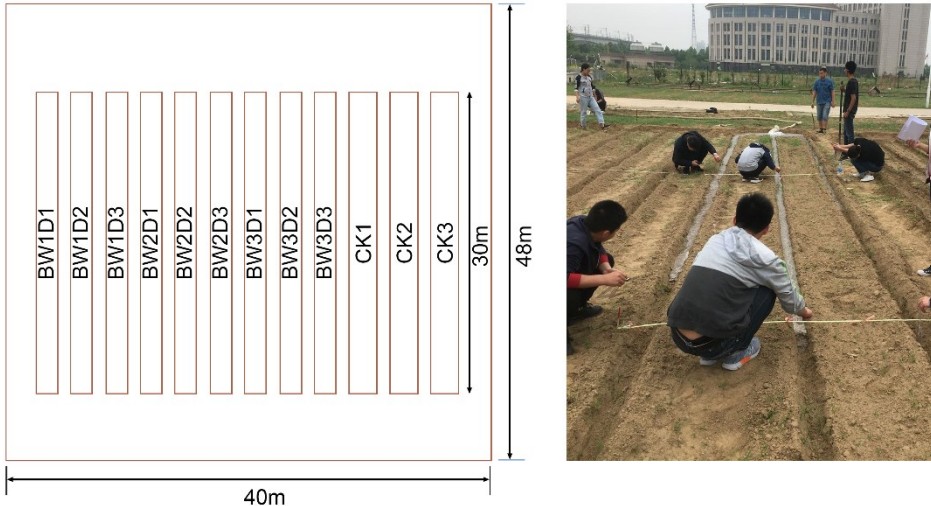

**Figure 1.** Details of experimental field.

### 2.2. Sampling and Measurements

Data collected for irrigation events was as follows: soil water content (*SWC*) before and after irrigation, inflow rate, advance and recession times of water, cut-off time, flow depth, and irrigation quota. The *SWC* was measured using an oven drying method. Before irrigation (24 h), a soil core sampler was used to collect soil samples from four randomly selected plots. Soil samples were collected to a soil depth of 60 cm from ground level (0–10 cm, 10–20 cm, 20–40 cm, and 40–60 cm layers). Each sample was placed into an aluminum box and weighed. The samples were then oven-dried at 105 °C for 24 h and weighed again. The *SWC* was calculated as follows:

$$SWC = \frac{m_f - m_d}{m_d} \times 100 \tag{1}$$

where *SWC* is the soil water content (%); $m_f$ is the fresh soil weight (g); $m_d$ is the dry soil weight (g).

After irrigation (48 h), soil samples were collected on the ridges and furrows at 5 m intervals along the micro-furrows for each treatment to analyze the soil water distribution and calculate the irrigation performance indicators. The sampling points for each profile are shown in Figure 2.

The advance time and recession time were determined at 5 m intervals in the longitudinal direction of the micro-furrow, along which we placed stakes every 5 m. The cut-off ratio was set as 1.0 for the 30 m-long micro-furrows, and the cut-off time was measured until the water flow advanced to the end of the micro-furrows. The irrigation quota was measured with a flow meter installed on the pipelines.

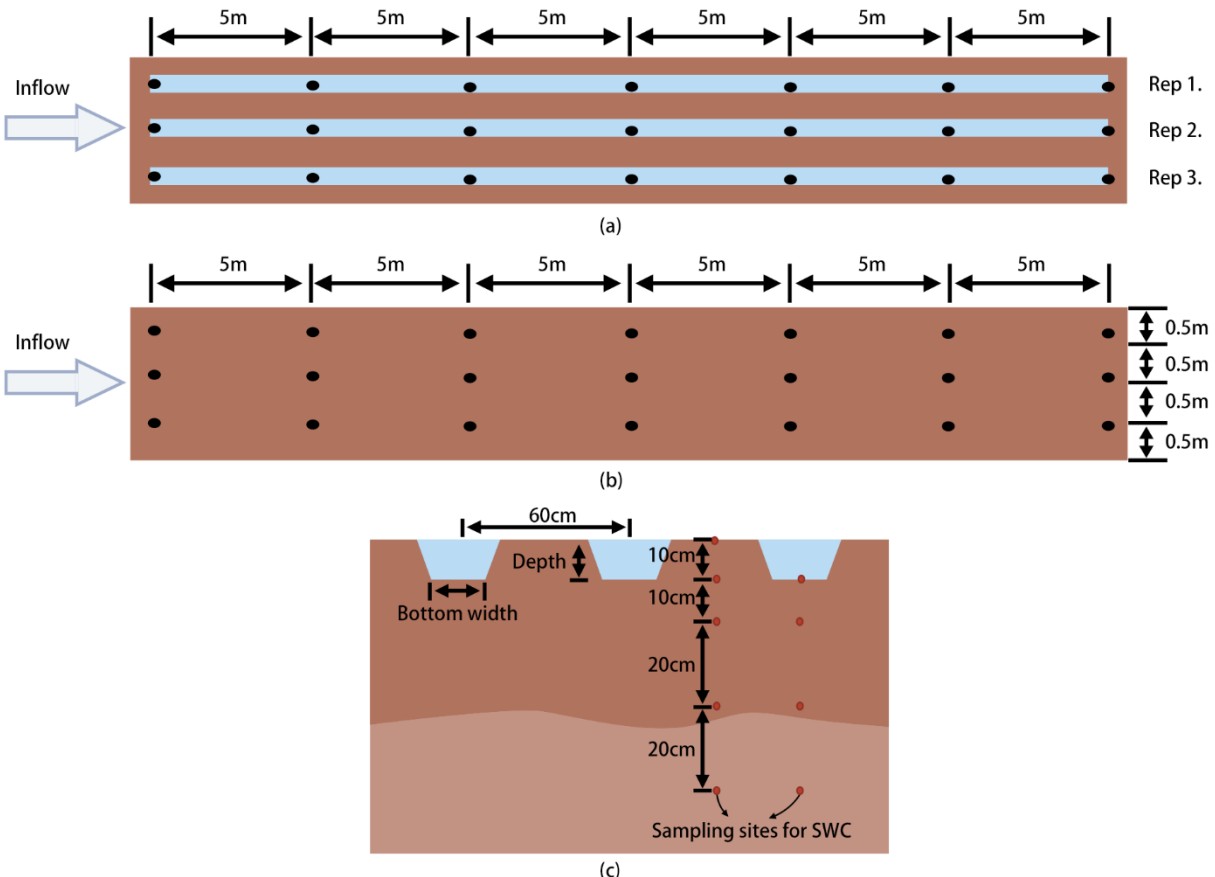

**Figure 2.** Illustration of the cross-section of the micro-furrows and sampling sites. (**a**) represent micro-furrow irrigation, (**b**) represent border irrigation, and (**c**) represent the cross-section of the micro-furrows.

### 2.3. Simulation Using WinSRFR5.1

WinSRFR is a software package for the hydraulic analysis of surface irrigation systems that was developed by the Agricultural Research Service of the United States Department of Agriculture. It offers four analytical functionalities in WinSRFR Worlds: Simulation, Event Analysis, Physical Design, and Operations Analysis. In this study, the advance time, recession time, and irrigation performance were simulated using WinSRFR5.1. Data required for simulating each irrigation event included system geometry, inflow rate, required depth, cut-off time, advance time, and recession time.

The two parameter Kostiakov infiltration equation was used to calculate the depth of infiltration in this study, which is expressed as follows:

$$z = Kt^{\alpha} \tag{2}$$

where $z$ is the depth of infiltration (mm), $K$ is the fitted coefficient parameter (mm/h$^{-1}$), $t$ is the infiltration time (h), and $\alpha$ is the fitted exponent parameter.

The value of the parameter $n$ was calculated as follows:

$$n = \frac{A^{5/3}S^{1/2}}{Q\chi^{2/3}} \tag{3}$$

where $n$ is Manning's roughness coefficient, $A$ is the cross section of the micro-furrow (m$^2$), $S$ is the field slope of the micro-furrow (m/m), $Q$ is the inflow rate (m$^3$/s), and $x$ is the wetted perimeter of micro-furrow (m).

The infiltration parameters $\alpha$ and $K$ were obtained in the WinSRFR software. In Event Analysis World, we selected Merriam–Keller post-irrigation volume balance analysis method, for which the available measurements were inflow hydrograph, advance time, and recession time. We also selected the zero-inertia model. We put advance time and recession time data coupled with $n$ into the WinSRFR software, and then we compared simulated advance and recession curves with the measured curves, until a good fit between simulated and measured curves was achieved, the combinations of $n$, $\alpha$ and $K$ were got. The parameters of infiltration and Manning's roughness coefficient are presented in Table 2.

**Table 2.** Parameters for infiltration and Manning's roughness coefficient.

| Treatments | $\alpha$ (−) | $K$ (mm/h$^{-1}$) | $n$ (−) |
|:---:|:---:|:---:|:---:|
| BW1D1 | 0.690 | 61.031 | 0.141 |
| BW1D2 | 0.390 | 87.810 | 0.074 |
| BW1D3 | 0.180 | 129.742 | 0.025 |
| BW2D1 | 0.510 | 72.311 | 0.094 |
| BW2D2 | 0.240 | 100.698 | 0.048 |
| BW2D3 | 0.250 | 67.427 | 0.016 |
| BW3D1 | 0.350 | 95.724 | 0.051 |
| BW3D2 | 0.380 | 70.114 | 0.024 |
| BW3D3 | 0.370 | 53.980 | 0.010 |
| CK | 0.290 | 66.792 | 0.040 |

### 2.4. Performance Evaluation of Micro-Furrow Irrigation

The irrigation performance was evaluated to determine whether the irrigation method met the water requirements and to ensure a uniform distribution of applied water in the field. Typically, the performance indicators for evaluating the irrigation performance include the application efficiency (*AE*), distribution uniformity (*DU*), deep percolation (*DP*), and requirement efficiency (*RE*). In this experiment, *AE*, *DU*, and *RE* were used to evaluate the irrigation performance. These indicators are defined as follows:

$$AE = \frac{D_{ad}}{D_{app}} \times 100 \tag{4}$$

$$DU = \left[ 1 - \frac{\sum_{i=1}^{N} \left| \theta_i - \bar{\theta} \right|}{N\bar{\theta}} \right] \times 100 \tag{5}$$

$$RE = \frac{D_{ad}}{D_{dw}} \times 100 \tag{6}$$

where $D_{ad}$ is the depth of water added to the root zone (mm), $D_{app}$ is the depth of water applied to the micro-furrow (mm), $\theta_i$ is the observed *SWC* for the $i$th grid (%), $\bar{\theta}$ is the mean *SWC*, $N$ is the number of grids, and $D_{dw}$ is the depth of water required for the micro-furrow (mm), which was considered as 40 mm for the field experiment in this study.

The comprehensive performance (*CP*) was calculated using Equation (7). Higher values of *CP* indicate better irrigation performance.

$$CP = aAE + bDU + bRE \tag{7}$$

where $a = b = c = 1/3$. We assumed that three indicators were equally important, so the weight values of $a$, $b$ and $c$ were the same.

Additionally, the spatial and profile distributions of soil water were used to analyze how the applied water was distributed under different treatments.

## 3. Results

### 3.1. Advance Time and Recession Time

The advance time for micro-furrow irrigation was shorter than that for border irrigation. There was a significant negative relationship between micro-furrow depth and advance time (Figure 3(a1,b1,c1)). Under the same bottom width, the lowest advance times were obtained under D1, and the highest were obtained under D3. The advance times were 3.91 min, 3.43 min, 1.39 min, 4.77 min, 2.88 min, 1.23 min, 4.59 min, 2.68 min, and 1.57 min lower for the BW1D1, BW1D2, BW1D3, BW2D1, BW2D2, BW2D3, BW3D1, BW3D2, and BW3D3 treatments, respectively, than that for the CK treatment. Therefore, micro-furrow irrigation can shorten the advance time compared with border irrigation. Moreover, the advance time increased as the depth of the micro-furrows decreased.

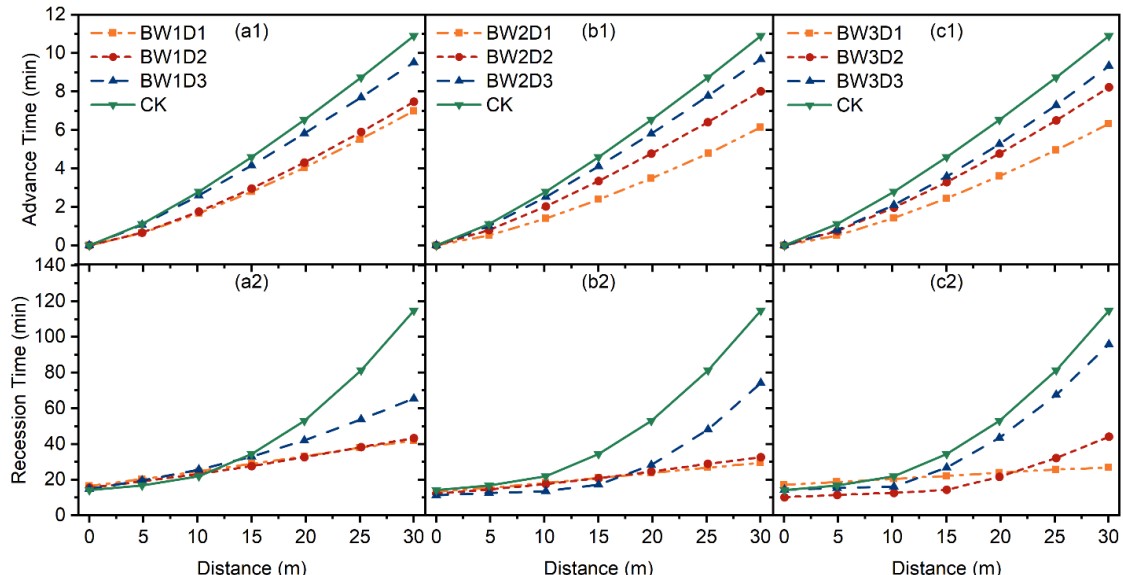

**Figure 3.** Advance and recession times along the micro-furrow length under different treatments. (**a1**), (**b1**), and (**c1**) represent the advance time, (**a2**), (**b2**), and (**c2**) represent the recession time.

Under the same depth, the advance times were similar under different bottom widths. This indicated that there were no significant differences between the advance time and bottom width. Furthermore, we found that a micro-furrow with a larger depth and smaller bottom width was conducive to accelerating the advance time.

The recession times along the micro-furrow length for different treatments are shown in Figure 3(a2,b2,c2). Under BW1D1 and BW1D2, and BW2D1 and BW2D2, the recession times were similar. As depth decreased from D1 to D3, the recession time increased by 23.53 min under BW1, 44.83 min under BW2, and 68.96 min under BW3. The lowest recession time was obtained under BW3D1. As in the relationship between micro-furrow depth and advance time, the recession time increased as the depth of the micro-furrows decreased. Additionally, recession time and advance time were directly related.

However, as bottom width decreased, the change in recession time varied under different depths. Under D1, the recession time decreased by 15.12 min and 2.6 min under BW3 compared with that under BW1 and BW2, respectively. Under D2, the recession time increased by 0.9 min and 11.48 min under BW3 compared with that under BW1 and BW2, respectively. Under D3, the recession time increased by 30.31 min and 21.53 min under BW3 compared with that under BW1 and BW2, respectively.

Significantly decreased recession times occurred under BW1D1, BW1D2, BW1D3, BW2D1, BW2D2, BW2D3, BW3D1, BW3D2, and BW3D3 compared with that under CK, with values of 72.75 min, 71.42 min, 49.22 min, 85.27 min, 82.22 min, 40.44 min, 87.87 min, 70.52 min, and 18.91 min, respectively. No significant difference was observed between

these results for D1 and D2. Therefore, our results showed that the recession times of micro-furrow were better than those of border irrigation.

### 3.2. Soil Water Distribution

#### 3.2.1. Spatial Distribution of Soil Water

The spatial distribution of soil water in the 0–60 cm soil layer under different treatments after two days of irrigation are shown in Figure 4. Before irrigation, the mean *SWC* in the 0–60 cm soil layer was 13.4%. After two days of irrigation, the *SWC* of this soil layer had increased. In the longitudinal direction, the *SWC* values under CK treatment in the front portion were 2.30% higher than those in the tail. However, under the micro-furrow irrigation treatments, the SWC values in the front portion of micro-furrow were higher than those in the tail by 1.64%, 0.58%, 0.05%, 1.35%, 0.78%, 0.70%, 1.02%, 2.45%, and 7.27% under BW1D1, BW1D2, BW1D3, BW2D1, BW2D2, BW2D3, BW3D1, BW3D2, and BW3D3, respectively. Except under BW3D2 and BW3D3, the spatial distribution of soil water in 0–60 cm soil layer was better than CK. The main reason was that the smaller bottom width and depth led to the collapse and overflow of the micro-furrows during irrigation.

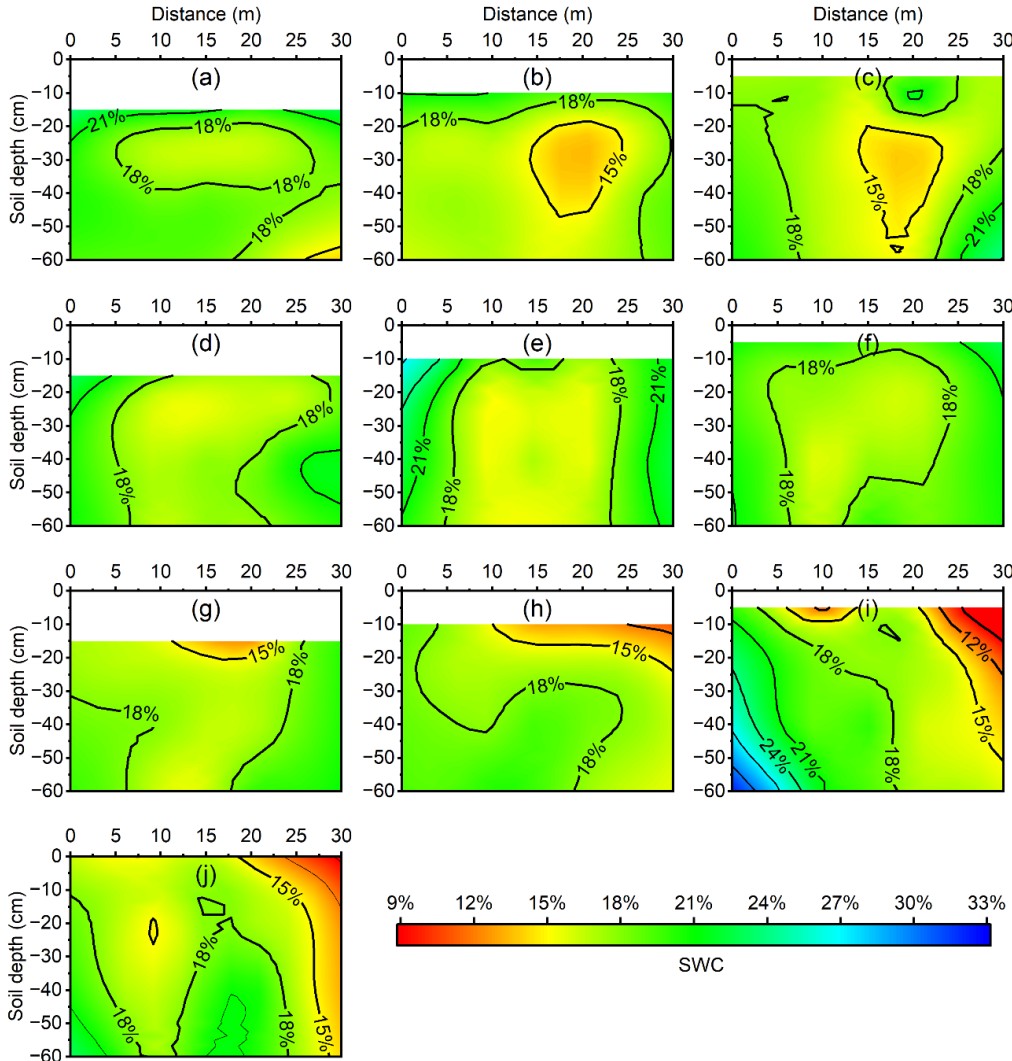

**Figure 4.** Spatial distribution of soil water in the 0–60 cm soil layer under different treatments after 2 days of irrigation. (**a**) BW1D1, (**b**) BW1D2, (**c**) BW1D3, (**d**) BW2D1, (**e**) BW2D2, (**f**) BW2D3, (**g**) BW3D1, (**h**) BW3D2, (**i**) BW3D3, and (**j**) CK.

The spatial distribution uniformity ($DU_S$) under different treatments is shown in Figure 5a. For the ten treatments, the $DU_S$ of micro-furrow irrigation was generally greater than that of border irrigation. Compared with those of D1, the $DU_S$ values under D2 and D3 were 2.43% and 2.99% lower under BW1, 4.21% and 0.23% lower under BW2, and 0.91% and 4.44% lower under BW3, respectively. Therefore, deep micro-furrows improved the $DU_S$. Additionally, we found that the $DU_S$ first increased and then decreased with decreasing bottom width.

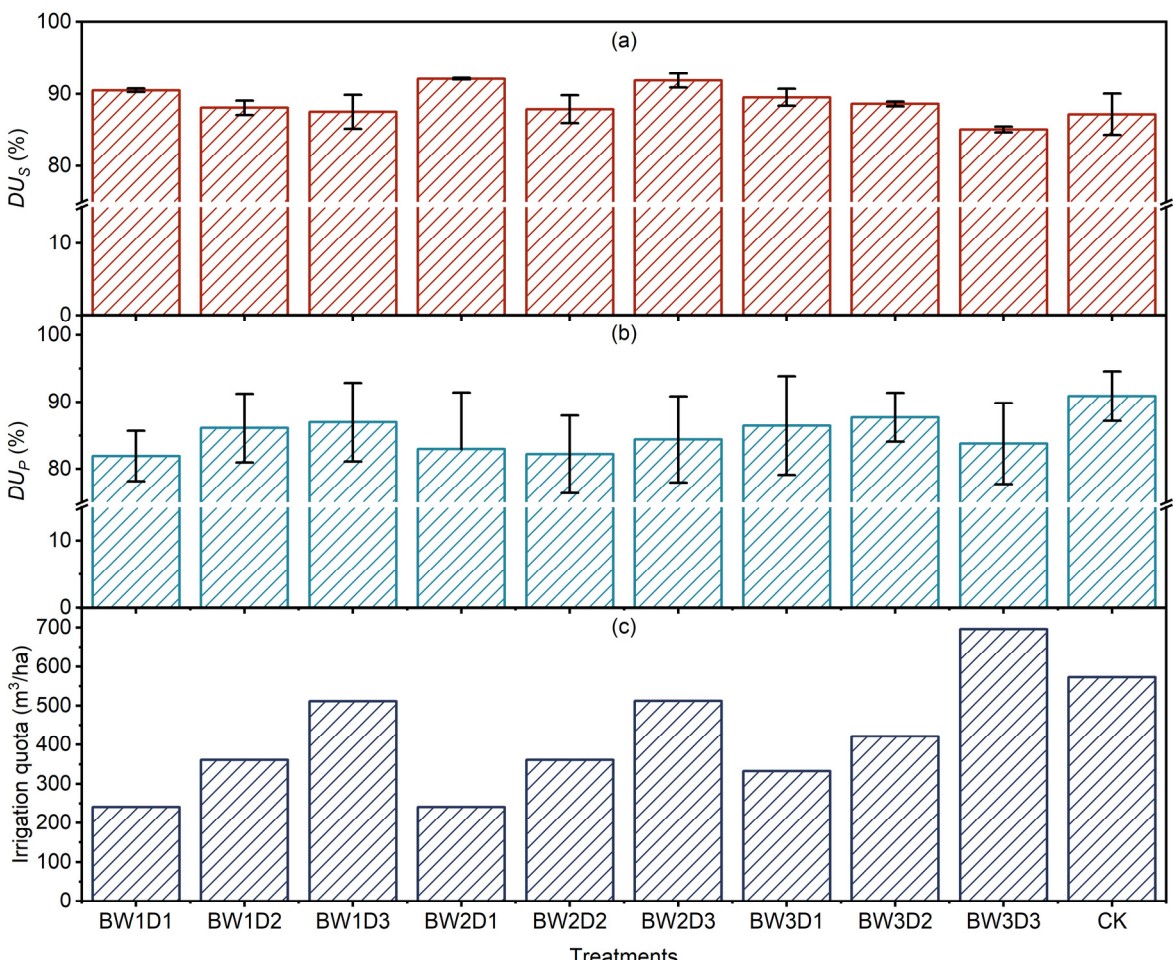

**Figure 5.** Bar graphs of the (**a**) spatial distribution uniformity, (**b**) profile distribution uniformity, and (**c**) irrigation quota under different treatments.

### 3.2.2. Profile Distribution of Soil Water

After 2 days of irrigation, compared with the CK treatment, the *SWC* values in the 0–10 cm layer on ridges were lower, whereas those in the 10–60 cm layer under ridges and micro-furrows were similar (Figure 6a–i). As shown in Figure 5b, the profile distribution uniformity ($DU_P$) values for micro-furrow irrigation were lower than those for border irrigation. This is mainly due to the lower *SWC* values in the 0–10 cm layer under the ridges. It is worth mentioning that the *SWC* values in the 10–60 cm layer under ridges and micro-furrows were not significantly different, which indicated good lateral infiltration.

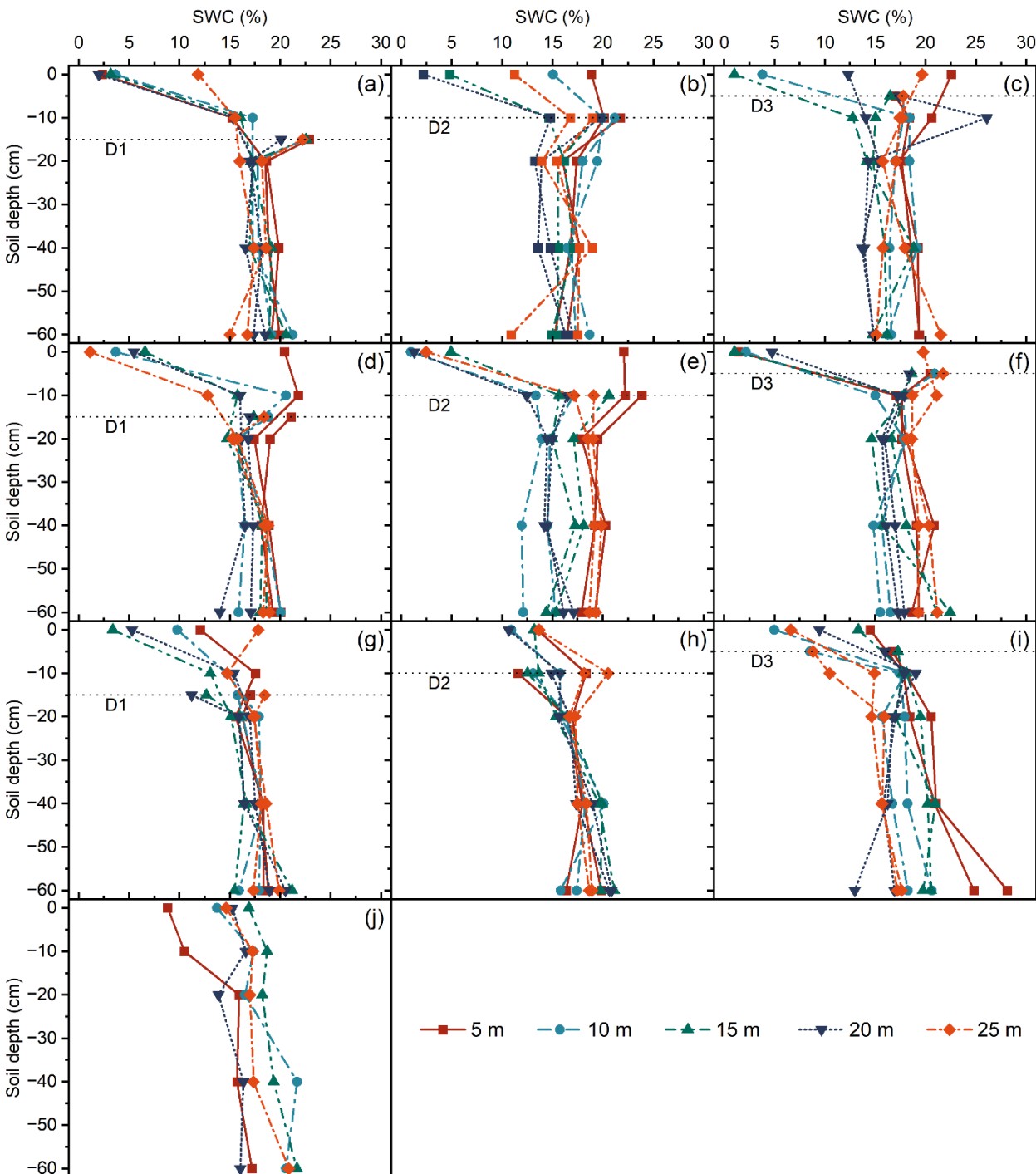

**Figure 6.** Profile distribution of soil water in different layers under different treatments after 2 days of irrigation. There are 2 series of value with same maker, line type and color. One series start from zero represents the *SWC* values on ridges, and the other from beneath represents the *SWC* values on micro-furrows. (**a**) BW1D1, (**b**) BW1D2, (**c**) BW1D3, (**d**) BW2D1, (**e**) BW2D2, (**f**) BW2D3, (**g**) BW3D1, (**h**) BW3D2, (**i**) BW3D3, and (**j**) CK.

### 3.3. Irrigation Quota

The irrigation quota under different treatments is shown in Figure 5c. There was a significant negative relationship between micro-furrow depth and irrigation quota. There was a significant difference in irrigation quota between BW1 and BW3, but no significant

difference was observed between that under BW1 and BW2. This was mainly related to advance time.

In our study, the irrigation quota decreased by 57.94%, 36.99%, 10.61%, 57.94%, 36.95%, 10.46%, 41.85%, and 26.34% under BW1D1, BW1D2, BW1D3, BW2D1, BW2D2, BW2D3, BW3D1, and BW3D2 treatments, respectively, compared with that under the CK treatment. However, the irrigation quota increased by 21.61% under BW3D3 compared with that under the CK treatment. Therefore, micro-furrow irrigation can be applied under a low irrigation quota.

### 3.4. Evaluation of Irrigation Performance

The evaluation of irrigation performance under different treatments is presented in Table 3. Under the nine treatments with micro-furrow irrigation, the *AE* ranged from 78% to 100.00%, *DU* ranged from 73% to 96%, and *RE* ranged from 60% to 100%. Compared with those under CK treatment, the *AE* and *DU* increased by 8–30% and −5–18%, respectively, whereas the *RE* decreased by 0–40%. Consequently, the *CP* values under all treatments except BW1D1 and BW2D1 were higher than those under the CK treatment. Our results indicated that micro-furrows improved irrigation performance. The three highest *CP* were obtained under BW2D2, BW3D1, and BW1D2.

**Table 3.** The evaluation of irrigation performance under different treatments.

| Treatments | *AE* (%) | *DU* (%) | *RE* (%) | *CP* (%) |
|---|---|---|---|---|
| BW1D1 | 100 | 73 | 60 | 76.89 |
| BW1D2 | 100 | 83 | 90 | 90.09 |
| BW1D3 | 78 | 89 | 100 | 88.11 |
| BW2D1 | 100 | 84 | 60 | 80.52 |
| BW2D2 | 100 | 92 | 90 | 93.06 |
| BW2D3 | 78 | 86 | 100 | 87.12 |
| BW3D1 | 100 | 96 | 83 | 92.07 |
| BW3D2 | 91 | 85 | 96 | 89.76 |
| BW3D3 | 88 | 77 | 91 | 84.48 |
| CK | 70 | 78 | 100 | 81.84 |

## 4. Discussion

### 4.1. Advance Time and Recession Time

The advance time and recession time are important indicators that affect the irrigation performance [42]. For surface irrigation, each point along the longitudinal direction in the irrigation unit does not begin and stop receiving water at the same time, resulting in varying opportunity times [53]. Long advance time and recession time lead to more infiltration in the front and less infiltration in the tail, resulting in deep percolation, poor distribution uniformity, and low water use efficiency [54]. In the present study, we found that the advance time and recession time of micro-furrow irrigation were lower than those of border irrigation. In general, the advance time of micro-furrow irrigation was 1.27–4.77 min shorter than that of the CK treatment, and the recession time was 18.91–87.87 min shorter than that of the CK treatment (Figure 3). Thus, micro-furrow irrigation reduces the difference in opportunity time at each point along the longitudinal direction and improves the movement of surface water and uniform distribution of soil water. In addition, when ponding occurs in a field, micro-furrows can be used to quickly remove water to avoid damage to crops.

In the present study, we also found that the advance time increased with decreasing micro-furrow depth. The largest advance times were obtained under D3. Prior studies indicated that field geometry, inflow rate, infiltration parameters and Manning's roughness significantly affected the advance time [42,44,53]. In our study, we controlled the field geometry, inflow rate, and field slope of all treatments to be consistent. Therefore, the differences in the advance time under different depths might be due to the changes in the infiltration parameters and Manning's roughness. Compared with that under D1 and D2,

the highest Manning's roughness value was under D3, which caused the greatest resistance to advancing water. Moreover, the highest infiltration parameter (*K*) occurred under D3, which reduced the driving force; thus, the largest infiltrated volume and cumulative infiltration were obtained under D3 [45]. Jin [55] conducted field and simulated furrow irrigation experiments to assess the effects of furrow bottom width on the advance time. The results showed that the advance time was closely related to the furrow bottom width. A smaller bottom width produced a smaller wet perimeter, which reduced the resistance to water flow. In contrast, in the present study, we found no significant difference between the advance time and bottom width; thus, our results differ from those of the previous research. This inconsistency was mainly due to the limited range of furrow bottom widths, which led to smaller variations in the wetted perimeter. Consequently, the advance time was almost unchanged with the decrease in bottom width. Similar to Xu et al. [42] and Mazarei et al. [45], our results showed that the recession time increased as the advance time increased.

*4.2. Soil Water Distribution and Evaluation of Irrigation Performance*

Soil water distribution is the visual embodiment of irrigation performance. In our study, we found that the spatial distribution of soil water in the 0–60 cm soil layer was better than that of the CK treatment, except BW3D2 and BW3D3. Considering the profile distribution of soil water, the *SWC* values in the 0–10 cm layer under micro-furrow irrigation were lower than those under CK, whereas the *SWC* values in the 10–60 cm layer were similar to those of CK. This was mainly due to the difference in water infiltration. The water from border irrigation enters the soil through vertical infiltration, which is conducive to a vertical profile distribution of soil water. For micro-furrow irrigation, the water enters the soil through vertical and lateral infiltration, which is conducive to horizontal spatial distribution of soil water. Therefore, compared with those of border irrigation, the $DU_P$ values of the micro-furrow irrigation were lower than that of border irrigation, the $DU_S$ values of micro-furrow irrigation were generally higher. We also found that the *SWC* values in the 10–60 cm layer under both of ridges and micro-furrows increased to the same extent. This indicated that the micro-furrow spacing was within the maximum lateral infiltration spacing and the water could infiltrate into the middle of the ridge. Nie et al. [56] showed that spacing has minimal impact on cumulative infiltration but has a large impact on the profile distribution of soil water. In other words, the intersection infiltration of the zero-flux surface occurs within the maximum lateral infiltration spacing, and the *SWC* at the same position of the zero-flux surface decreases with increasing furrow spacing. In this study, only one micro-furrow spacing was used. Hence, the optimal infiltration spacing was not clear. Therefore, the spacing needs to be examined in a future study.

The common performance indicators for evaluating irrigation performance are *AE*, *DU*, and *RE* [18,25,30,57]. Ideal irrigation performance maximizes all three values. However, this is difficult to realize in practice because *AE* and *RE* contradict each other. Liu et al. [58] indicated that *DU* is the key for improve irrigation performance. This view supports our study. In our study, we compared the performance indicators of micro-furrow irrigation and border irrigation among the ten treatments. Although the values of *RE* decreased, the values of *AE* and *DU* increased (Table 3). The above analysis shows that micro-furrow irrigation improves irrigation performance.

*4.3. Irrigation Quota*

Irrigation quota is an important factor for determining whether an optimized irrigation schedule can be applied. At present, China implements a very stringent water resources management system, which controls the total amount of water used. Under these conditions, the corresponding irrigation schedule must be adjusted to ensure an increasing and stable yield of crops. Various studies have adjusted irrigation time and irrigation quota to maximize crop yields [59–63]. However, an optimized irrigation schedule is mostly suitable for pressure irrigation systems, such as sprinkler irrigation, drip irrigation, and

micro sprinkler irrigation. For surface irrigation, the actual irrigation quotas are larger than the designed irrigation quotas due to long advance times [64]. Previous studies attempted to apply precision land levelling applied [65,66]. In our study, the irrigation quotas in all treatments except BW3D3 were lower than that of the CK treatment (Figure 5c). Thus, our results indicated that micro-furrow irrigation can be applied under a low irrigation quota.

## 5. Conclusions

In this study, we proposed a technology, namely micro-furrow irrigation, which combines the advantages of furrow irrigation and border irrigation. The bottom width and depth of the micro-furrows were evaluated to analyze their effects on surface water flow and improve irrigation performance. There was a significant negative correlation between the micro-furrow depth and advance time, but bottom width and advance time were not significantly correlated. Additionally, recession time and advance time were directly related. Although deep micro-furrows reduced the $DU_P$, they improved the $DU_S$. Our study examined the *AE*, *DU*, *RE*, and *CP* under nine micro-furrow irrigation treatments. The *AE* ranged from 78% to 100.00%, the *DU* ranged from 73% to 96%, and the *RE* ranged from 60% to 100%. The three highest *CP* were obtained under BW2D2, BW3D1, and BW1D2. Compared with border irrigation, the advance time and recession time of micro-furrow irrigation were lower, the $DU_S$ values were generally higher, and the $DU_P$ values and irrigation quota were lower. The *AE* and *DU* increased, whereas the *RE* decreased. Our results indicated that micro-furrow irrigation can meet irrigation requirements under a low irrigation quota. Overall, the micro-furrow irrigation technology can shorten the advance time and recession time of surface water flow and improve irrigation performance in the North China Plain. Comprehensively considering the advance time and recession time, irrigation performance, and irrigation quota, we recommend a micro-furrow shape with a depth of 10 cm or 15 cm and a bottom width of 6 cm.

**Author Contributions:** Conceptualization, S.Z. and F.W.; methodology, S.Z., X.F. and F.W.; software, X.F.; validation, S.Z. and X.F.; formal analysis and investigation, S.Z., X.F., F.W. and Y.Z.; resources, S.Z. and F.W.; data curation and writing—original draft, S.Z., X.F., D.W. and F.W.; writing—review and editing, S.Z., X.F., D.W. and F.W.; visualization, S.Z., X.F. and F.W.; supervision, S.Z. and F.W.; project administration, S.Z. and F.W.; funding acquisition, S.Z. and F.W. All authors have read and agreed to the published version of the manuscript.

**Funding:** This research was supported by the National Natural Science Foundation of China (51979108), the Zhongyuan Science and Technology Innovation Leadership Project (194200510008), National Key Research and Development Program of China (2016YFC0400103) and the Innovation Funds of NCWU for Doctoral Candidate (B2020081505).

**Institutional Review Board Statement:** Not applicable.

**Informed Consent Statement:** Not applicable.

**Data Availability Statement:** Data recorded in the current study are available in all tables and figures of the manuscript.

**Acknowledgments:** We would like to thank the editor and the expert reviewers for their detailed comments and suggestion for the manuscript. These were very useful to hopefully improve the quality of the manuscript.

**Conflicts of Interest:** The authors declare no conflict of interest.

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
