# Peer review of "Influence of Micro-Furrow Depth and Bottom Width on Surface Water Flow and Irrigation Performance in the North China Plain"

_agronomy, doi:10.3390/agronomy12092156_

Round 1

Reviewer 1 Report

1.       A very interesting research, proposing an alternative irrigation method to increase efficiency. The paper is more at irrigation engineering, rather than agronomy science.

2.       The dimension of the land was mentioned 48x40 m2, however the schematics of experiment and observation as in figrure 1 are only having length of 30m (6x5m) which is a little bit confusing. The 48x40 m2 field schematic will be nice.

3.       Figure 5, a-i are confusing. Why there are 2 series of value with same marker, line type and color? For example brown box marker and solid line. One series start from zero and the other from beneath. What are the differences? Adding separate legends of data series will be helpful.

4.       How did you obtain Dad for calculationn of AE?

5.       Why a=b=c in eq(7)?

Reviewer 2 Report

Hi Authors,

Thank you for your interesting potential work. Each work needs some additional work to complete the scientific story for the readers.

Introduction:

Line 43-44 "More than 1.2 billion people in agricultural

 areas worldwide face serious water resource pressure and drought." this needs a reference to support what you said.

Line 63. You have to talk about the new innovative technology in irrigation methods to improve irrigation efficiency. Here need more references: 

https://www.sciencedirect.com/science/article/pii/S0378377421007137?casa_token=IRCnrE1SvjsAAAAA:CTCQTG287W2YI4_yq9uDc-NcQwD2W3BUhpyeVRo_J65e17mfnF7XxRDlPqV53ci11kL8KMHEy0I;

https://www.mdpi.com/2077-0472/11/2/87; https://www.mdpi.com/2073-4395/11/12/2493; https://www.mdpi.com/2073-4395/12/5/1077.

Line 66-69."Compared with other irrigation methods, surface irrigation remains widely used worldwide because of its simple field engineering facilities, easy implementation, low energy consumption, and low cost. It services approximately 90% of the total irrigated lands worldwide [10]" I am not sure this information is right.  

Please check this reference https://www.mdpi.com/2071-1050/13/8/4328;

and Mateos, L. Irrigation Systems BT—Principles of Agronomy for Sustainable Agriculture; Villalobos, F.J., Fereres, E., Eds.; Springer

International Publishing: Cham, Switzerland, 2016; pp. 255–267.

Line 103-105. "Therefore, a field experiment was conducted in the test field at Henan Key Laboratory of Water Saving Agriculture, and a simulated experiment was performed using the WinSRFR 5.1 model." this has to move to the Methodology section.

Materials and Methods:

You have to put your real picture and field setup for your new irrigation design. you need to insert a picture after line 122 

Line 130. "No crops were planted in the experimental field during the experiments." I don't why you did not perform this experiment with the plant to check how irrigation improves and whether water application covers plants' needs or not.

Fig (1). you have to insert a picture of the real field.

Equation 2. you have to maintain what these parameters are related with "K is the fitted coefficient parameter and the fitted exponent parameter and how can solve them.

Table 2. you have to make all the numbers the same after the dot for "k' and "n" such as 61.031 become 61.03 For 'K' and 0.141 become 0.14 for 'n'.

 Good luck
